# Stimulus-specific plasticity in human visual gamma-band activity and functional connectivity

**Benjamin J Stauch[1,2,3]\*, Alina Peter[1,2], Heike Schuler[1], Pascal Fries[1,2,3,4]\***

[1]Ernst Strüngmann Institute (ESI) for Neuroscience in Cooperation with Max Planck Society, Frankfurt, Germany; [2]International Max Planck Research School for Neural Circuits, Frankfurt, Germany; [3]Brain Imaging Center, Goethe University Frankfurt, Frankfurt, Germany; [4]Donders Institute for Brain, Cognition and Behaviour, Radboud University Nijmegen, Nijmegen, Netherlands

**Abstract** Under natural conditions, the visual system often sees a given input repeatedly. This provides an opportunity to optimize processing of the repeated stimuli. Stimulus repetition has been shown to strongly modulate neuronal-gamma band synchronization, yet crucial questions remained open. Here we used magnetoencephalography in 30 human subjects and find that gamma decreases across ≈10 repetitions and then increases across further repetitions, revealing plastic changes of the activated neuronal circuits. Crucially, increases induced by one stimulus did not affect responses to other stimuli, demonstrating stimulus specificity. Changes partially persisted when the inducing stimulus was repeated after 25 minutes of intervening stimuli. They were strongest in early visual cortex and increased interareal feedforward influences. Our results suggest that early visual cortex gamma synchronization enables adaptive neuronal processing of recurring stimuli. These and previously reported changes might be due to an interaction of oscillatory dynamics with established synaptic plasticity mechanisms.

\*For correspondence:
benjamin.stauch@esi-frankfurt.de
(BJS);
pascal.fries@esi-frankfurt.de (PF)

## Introduction

While moving through natural environments, organisms rarely encounter random and temporally independent visual inputs. Instead, they see environment-specific stimuli and stimulus categories repeatedly. A specific environment comes with its own distribution of probable edge orientations, object categories, and visual image statistics in general (*Torralba and Oliva, 2003*), and as organisms spend extended periods in the same environment, their visual input is likely to be autocorrelated (*Dong and Atick, 1995*) and self-repeating (*Wilming et al., 2013*).

This input repetition presents an opportunity: If an organism manages to tune its input processing to the input it is presented with within short timescales, it will be able to process probable future inputs optimally. Several theories have been formulated on algorithms the visual system might use to achieve such tuning to the input statistics in the long run (*Olshausen and Field, 1996*; *Rao and Ballard, 1999*), but the specific implementations, as well as changes in input processing over short to medium timescales, are still a matter of active inquiry.

Stimulus repetition has been shown to lead to a reduction of firing rates in stimulus-driven neurons (*Desimone, 1996*; *Li et al., 1993*) and a decreased hemodynamic response (*Grill-Spector et al., 2006*; *Stern et al., 1996*) in visual areas, a phenomenon generally called *repetition suppression*. Importantly, this decrease of neuronal activity does not lead to decreases in detection performance. Instead, detection performance generally stays stable or even improves over stimulus repetitions (*Fiorentini and Berardi, 1980*; *Grill-Spector et al., 2006*).

But how does the brain keep or improve behavioral performance with less neuronal activity? Potentially, repetition suppression might specifically target neurons coding for redundant, already predicted, information (*Auksztulewicz and Friston, 2016*). Alternatively, behavior might rely primarily on the neurons most responsive to the repeated input, which might be exempted from repetition suppression (*Desimone, 1996*; *Homann et al., 2017*) or might even undergo repetition enhancement (*Lim et al., 2015*). Consistent with the latter, a further possibility is that the remaining, non-suppressed neurons fire more synchronously, effectively compensating for decreased firing rates via increased temporal overlap between action potentials (*Gotts et al., 2012*).

In area V1, such an increase in synchronous neuronal firing and oscillatory power in the gamma band has been reported (*Brunet et al., 2014*). Specifically, gamma-band power in the local field potential increased with the logarithm of the number of repetitions, accompanied by increased V1–V4 coherence and gamma spike-field locking in V4.

However, several questions remain open: (1) Are the changes in the neuronal circuits that underlie the observed gamma-power increase specific to the stimulus that induced them, or do they equally affect the processing of other stimuli? (2) Do gamma enhancements persist over a time frame of several minutes and the intervening presentation of other stimuli, or do they vanish quickly? (3) Does repetition-related gamma enhancement exist in humans and for untrained, novel stimuli?

In this study, we recorded source-localized (*Gross et al., 2001*; *Van Veen et al., 1997*) MEG in 30 participants while they were presented with a continuous sequence of repeated oriented gratings. We found that the repetition-related gamma enhancement effect is clearly present in humans, is stimulus specific, and persists over time and deadaptation.

## Results

Stimuli as well as trial and session structure are illustrated in *Figure 1*. In short, subjects initiated each trial by fixating a central fixation spot. After a baseline (1 s), a central static grating with one of four possible orientations (22.5°, 67.5°, 112.5°, or 157.5° from the horizontal) was shown. After a period of 0.3–2 s, the grating changed orientation by up to 0.9 degrees, while decreasing in

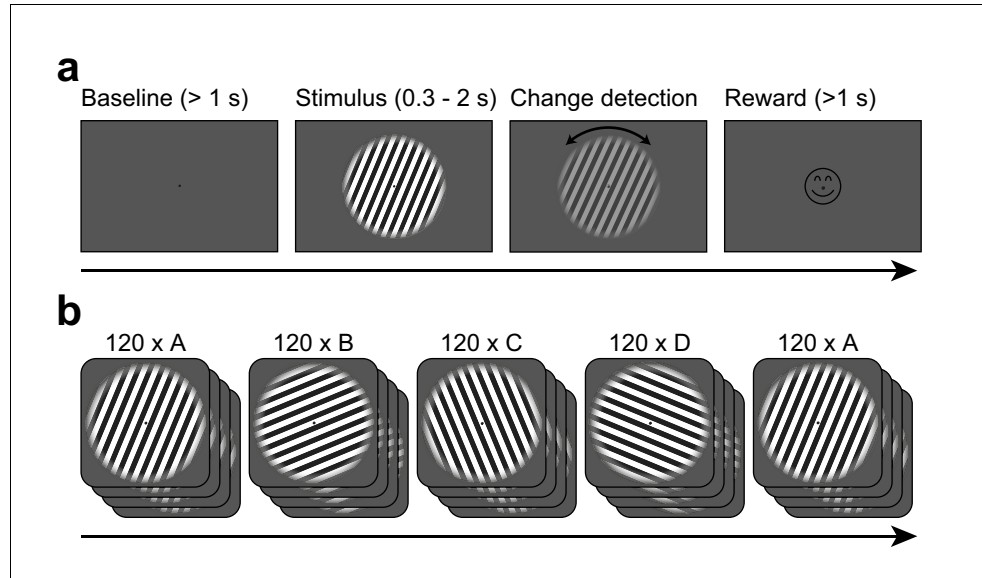

**Figure 1.** Task design. (a) Each trial started when gaze fixation was attained. A gray background was shown as a 1 s baseline, followed by a central grating (diameter = 22.9 deg). Between 0.3 and 2.0 s after grating onset, a contrast decrement and a small rotation were applied to the grating. The subject needed to report the rotation direction using a button press. Upon button press, a smiley was shown regardless of accuracy. Afterwards, a new trial was initiated. (b) The per-trial grating orientation followed a blocked ABCDA-pattern: 120 trials each of one of four possible orientations were shown, followed by another 120 trials of the orientation shown in the first block. There was no break or change of any kind between the blocks.

contrast. Subjects were required to report the direction of the orientation change. Each grating orientation was repeated for 120 trials in a blocked fashion (blocks A–D). After those blocks, the oriented grating of the first block was repeated again for another 120 trials (block A2). Except for the change in grating orientation, there was no change or break between the blocks.

To investigate how behavioral and neuronal responses ('responses of interest', e.g., gamma power, event-related field amplitude/ERFs) were affected by stimulus repetition, while controlling for other factors, we fitted separate random intercept linear regression models to each response of interest over all subjects. Each used the same independent variables (stimulus-specific repetition number, general trial number, microsaccade rate, and further covariates, see Materials and methods). As several of these responses showed different trajectories over repetitions 1–10 versus over all repetitions (e.g., an early decrease and an overall increase), we fitted two overlapping predictors for repetitions 1–10 and repetitions 1–120.

To analyze how early and late changes in different behavioral and neuronal responses correlated with each other, we fitted per-subject linear regression models to the responses of interest (separately for repetitions 1–10 and 11–120, as overlapping trajectories cannot be disentangled on a per-subject basis), using the same independent variables as above. Subsequently, the per-subject repetition-number coefficients were correlated between the behavioral and neuronal responses of interest.

## Subjects show valid, stable behavior

Subjects were able to distinguish the orientation change direction with a mean reaction time of 484 ms ($CI_{95\%} = [461\,\text{ms}\,510\,\text{ms}]$, all confidence intervals based on bootstrap procedures) and an above-chance accuracy of 69% ($CI_{95\%} = [63\%\,74\%]$, $p<4*10^{-7}$). Accuracy was not modulated by stimulus repetition number, total trial number, the repetition block, or the beginning of a new block (all $p>0.05$). By contrast, reaction times sped up by 15 ms over the first 10 presentations of an orientation block ($CI_{95\%} = [6\,\text{ms}\,24\,\text{ms}]$, $p<2*10^{-3}$) and then showed a small slowing of 0.1 ms per stimulus repetition ($CI_{95\%} = [0.05\,\text{ms}\,0.20\,\text{ms}]$, $p<2*10^{-3}$). The effects of total trial number and the repeat block (A2) on reaction times were small: A speed increase of –0.07 ms/trial over the whole experiment ($CI_{95\%} = [-0.09\,\text{ms} - 0.05\,\text{ms}]$, $p<2*10^{-13}$), and slower reaction times of 12 ms during the repeat block ($CI_{95\%} = [3\,\text{ms}\,19\,\text{ms}]$, $p<5*10^{-3}$). Changes in reaction times and accuracy were not correlated to changes in gamma power over subjects (see below).

## Stimuli induce gamma responses in visual areas

As expected, grating stimuli produced robust responses in visual areas: Dipoles in V1/V2 showed a clear visual ERF and a stimulus-driven gamma-band response (*Figure 2a–d*, *Figure 2—figure supplement 1c,d*). The gamma-band response was strongest in areas V1 and V2 and extended into temporal and parietal lobes (*Figure 2e*). Furthermore, a stimulus-driven decrease in source-localized alpha and beta power could be seen in temporal/parietal and parietal/frontal areas, respectively (*Figure 2—figure supplement 1c*). Frequency bands were determined (*Haller et al., 2018*) based on subject-individual spectra of stimulus-induced power changes, if possible (see Materials and methods for details).

## Stimulus repetition induces early decreases and later increases in gamma power that are both stimulus specific

The strength of the gamma-band response (measured as gamma power during stimulation/gamma power during trial-mean baseline) changed across repeated presentations of the same stimulus (*Figure 3*). Across stimulus repetitions, gamma showed a biphasic pattern that could be well described by a linear decrease until a breakpoint at the 10th repetition, followed by a linear increase (see Materials and methods). Across the first 10 stimulus repetitions after a stimulus-block start, gamma dropped by 30.8 pp (percentage points) ($CI_{95\%} = [22.4\,\text{pp}\,38.75\,\text{pp}]$, $p<2*10^{-14}$); we will refer to this as the early gamma-power decrease. In addition, over all repetitions, gamma continually increased with repetitions by about 0.40 pp/repetition of a specific stimulus ($CI_{95\%} = [0.33\,\text{pp}\,0.46\,\text{pp}]$, $p<2*10^{-16}$), which corresponds to an average increase of 48 pp over the 120 presented repetitions; we will refer to this as the gamma-power increase. We also fitted the pattern of gamma over all repetitions with the sum of an exponential decay and a linear increase and

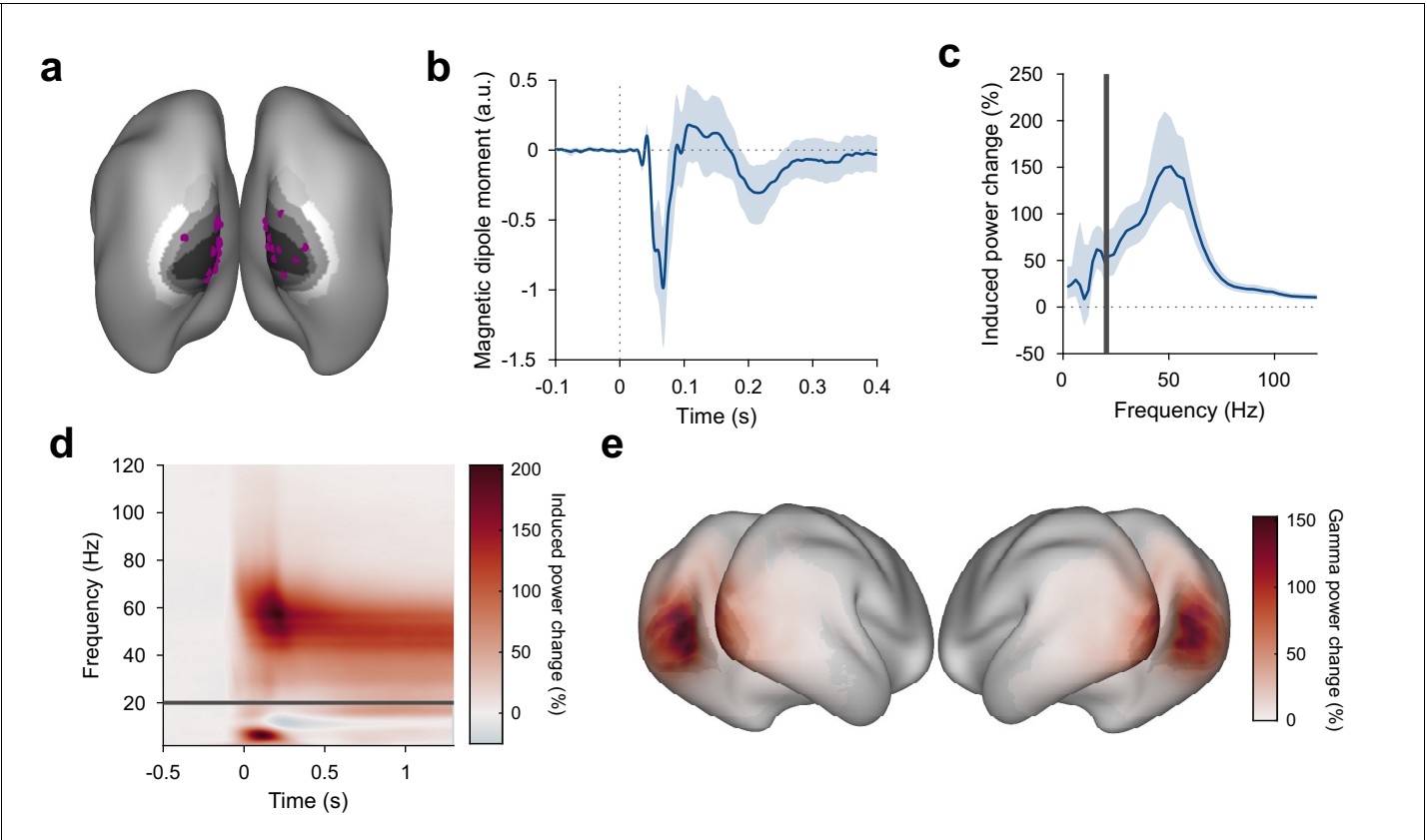

**Figure 2.** Stimulus-induced ERF and gamma-band response in visual cortex. (a) Each violet dot shows the selected dipole with the strongest visually induced gamma of one subject. Black-to-white shading indicates areas V1, V2, V3, V3A, and V4. All selected dipoles were located in areas V1 or V2. All analyses referring to activity in V1/V2 used the MEG data projected into these dipoles. (b) Average V1/V2 magnetic dipole moment in response to stimulus onset. (c) Average stimulus-induced power change in V1/V2, calculated as per-trial power from 0.3 to 1.3 s post-stimulus divided by average power during the 1 s baseline. Error bars in (b, c) show 95% confidence intervals based on a bootstrap across subjects. (d) Average stimulus-induced power change in V1/V2 as a function of time and frequency. In (c, d), power values from 1 to 20 Hz (below the gray bar) were computed using Hann tapering, power values of higher frequencies were computed using multi-tapering and line noise was removed using DFT filters. (e) Average stimulus-induced gamma-power change (individual gamma peak ±10 Hz), source projected to all cortical dipoles. Values are significance-masked using a $t_{max}$-corrected permutation test. Black-to-white shading indicates areas V1, V2, V3, V3A, and V4.

The online version of this article includes the following figure supplement(s) for figure 2:

**Figure supplement 1.** Stimulus-induced power changes.

found the early decrease to have a time constant of 3.5 repetitions ($CI_{95\%} = [2.3\,4.6]$). When the visual stimulus was switched at the beginning of a new block, this pattern of early decrease and subsequent increase repeated. This demonstrates that these repetition-related gamma increases were specific to the repeated stimulus because block boundaries were only constituted by switches in stimulus orientation.

In addition, we observed a stimulus-unspecific effect of trial number: The strength of the gamma-band response increased with total trial number by about 0.07 pp/total trial number ($CI_{95\%} = [0.06\,\mathrm{pp}\,0.09\,\mathrm{pp}]$, $p<2*10^{-16}$), which corresponds to an average increase of 44 pp over the total 600 trials of the experiment.

Furthermore, the gamma-power enhancement across stimulus repetitions partially persisted over more than 25 min of intervening presentation of other orientations: Induced gamma power during block A2 was on average a further 7.80 pp above the level predicted by all other factors (including total trial number, $CI_{95\%} = [0.90\,\mathrm{pp}\,14.87\,\mathrm{pp}]$, $p = 0.024$, *Figure 3c*).

Both the early decrease in gamma power and the gamma-power increase source-localized to visual cortical areas and were strongest in V1, V2, V3, and V4 (*Figure 3e,f*; for sensor-level analysis, see *Figure 2—figure supplement 1c* and *Figure 3—figure supplement 1e,f*). The repetition-

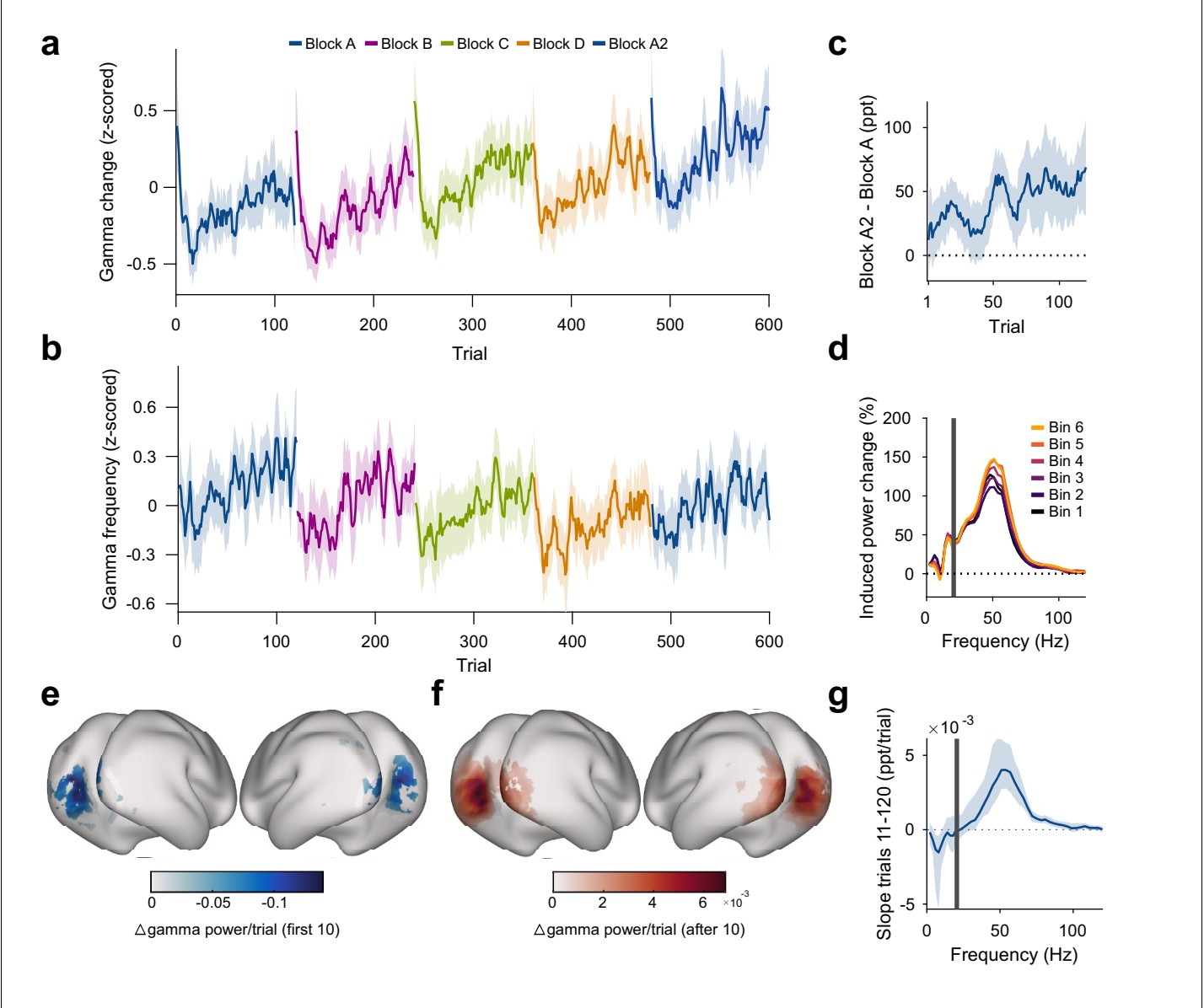

**Figure 3.** Repetition effects on gamma power and peak frequency are stimulus specific. (a) Stimulus-induced gamma power in V1/V2, on a per-trial basis. (b) Peak frequency of stimulus-induced gamma in V1/V2, on a per-trial basis. Values in (a, b) were z-scored within subjects. (c) Within-subject differences in stimulus-induced V1/V2 gamma power between the second and the first block of a given oriented grating (A2 minus A). Note that induced gamma power also showed an increase with stimulus-independent trial number, which is controlled for in the regression model presented in Results. In (a–c), the average and the 95% bootstrap confidence intervals were computed using a five-trial-wide running window. (d) Stimulus-induced power-change spectra in V1/V2 during the 120 presentations of a given stimulus, plotted in sequential 20-presentation bins. Power values from 1 to 20 Hz (left of the gray bar) were computed using Hann tapering, power values of higher frequencies were computed using multi-tapering. Line noise was removed using DFT filters. (e) Spatial distribution of the early gamma power decrease: For each cortical dipole, a regression line was fit to induced gamma power as a function of stimulus repetitions 1–10. Subject-averaged slopes (significance-masked, $t_{max}$-corrected) are shown. (f) Spatial distribution of the late gamma increase: For each cortical dipole, a regression line was fit to induced gamma power as a function of stimulus repetitions 11–120. Subject-averaged slopes (significance-masked, $t_{max}$-corrected) are shown. In (e, f), black-to-white shading indicates areas V1, V2, V3, V3A, and V4. (g) For each frequency, a linear regression across repetitions was fit to the per-trial visually induced power change in V1/V2 during the late trials (trials 11–120). Average slope and 95% bootstrap CI over subjects is shown. The corresponding analysis for the early trials (trials 1–10) is shown in *Figure 3—figure supplement 1d*. All results in this figure show averages over all participants.

The online version of this article includes the following figure supplement(s) for figure 3:

**Figure supplement 1.** Control analyses of repetition-induced gamma changes.

**Figure supplement 2.** Full regression model of gamma power and peak-frequency changes.

*Figure 3 continued on next page*

*Figure 3 continued*

**Figure supplement 3.** Per-trial changes in baseline alpha power and blink number.

related increase was specific to the gamma band (*Figure 3d,g*). Furthermore, it was specific to the trial epoch with visually induced gamma: Power in the gamma-band during the pre-stimulus baseline did not show an association with stimulus repetition number ($p = 0.36$, *Figure 3—figure supplement 1a*).

We controlled for changes in the rate of microsaccades (MSs). The MS rate had been included as a covariate in the gamma-power regression, which revealed that a higher MS rate was not significantly associated with stronger gamma power ($p = 0.17$). Furthermore, MS rate did not change with stimulus repetition number ($p = 0.66$) and slightly decreased with total trial number by 0.0004 sac/s/trial ($CI_{95\%} = [-0.0005 - 0.0002]$, $p < 2 * 10^{-3}$, *Figure 3—figure supplement 1b*). These observations together show that the gamma-power increase could not have been driven by changes in MS rate.

The per-subject magnitude of changes in gamma power over both the first 10 stimulus repetitions and over later stimulus repetitions was related neither to the per-subject magnitude of changes in accuracy over early or late stimulus repetitions nor to the per-subject magnitude of changes in reaction time over early or late stimulus repetitions (all $p > 0.22$).

## Gamma frequency mirrors gamma-power increase, but shows no early decrease

The repetition of a given stimulus affected not only gamma power but also gamma peak frequency (*Figure 3b*, determined per-subject, per-trial). Gamma peak frequency increased with stimulus repetitions by 0.05 Hz/repetition of a specific stimulus ($CI_{95\%} = [0.04\,\mathrm{Hz}\,0.06\,\mathrm{Hz}]$, $p < 2 * 10^{-16}$), which corresponds to an average increase of 6 Hz over the 120 presented repetitions. The first 10 repetitions, which had shown a distinct decrease for gamma power, did not show any significant changes for gamma peak frequency ($p = 0.19$). In addition, we observed a stimulus-unspecific effect of trial number, in which the gamma peak frequency decreased with trial number by about 0.01 Hz/total trial number ($CI_{95\%} = [0.008\,\mathrm{Hz}\,0.013\,\mathrm{Hz}]$, $p < 4 * 10^{-15}$), which corresponds to an average decrease of 6 Hz over the 600 total trials of the experiment. The gamma peak frequency increase over stimulus repetitions partially persisted from block A to A2: Gamma peak frequency during block A2 was a further 3.2 Hz above the level predicted by all other factors ($CI_{95\%} = [2.10\,\mathrm{Hz}\,4.32\,\mathrm{Hz}]$, $p < 2 * 10^{-8}$).

## Pupil constriction shows early decrease and then stabilizes

The switch of stimuli between blocks might have induced a change in arousal and pupil size. With the stimuli used here, stimulus presentations led to reliable pupil constrictions, as induced by the pupillary light reflex. Pupil constriction (the difference between pupil size before stimulus onset and 0.5 s–1.2 s after stimulus onset, *Figure 4b*, *Figure 4—figure supplement 1a*) decreased over the first 10 repetitions ($p < 3 * 10^{-10}$), but was not influenced by further stimulus repetitions ($p = 0.68$, *Figure 4b,c*) nor total trial number ($p = 0.64$). As for gamma, we also fitted the pattern of pupil constriction over all repetitions with the sum of an exponential decay and a linear increase and found the early decrease to have a time constant of 5.6 repetitions ($CI_{95\%} = [2.6\,8.7]$). The per-subject changes in pupil constriction (averaged over blocks) were correlated to the per-subject changes in induced gamma power (averaged over blocks) with stimulus repetition over the first 10 repetitions of each stimulus, i.e. during the early gamma-power decrease ($r_{Spearman} = 0.45, p = 0.013$), but not over all repetitions, that is during the gamma-power increase ($p = 0.20$). Thus, during the first 10 repetitions, across subjects, larger reductions in pupil constriction were accompanied by larger reductions in gamma.

The changes in pupil constriction over repetitions were not driven by changes in pre-stimulus pupil size: Pre-stimulus pupil size did not change over the first 10 stimulus repetitions ($p = 0.15$) nor with further stimulus repetitions ($p = 0.59$, *Figure 3—figure supplement 1c*).

## Event-related fields show slow stimulus-specific decreases

Source-reconstructed event-related fields (ERFs) in V1/V2 showed changes similar to the later gamma-power increase, but opposite in sign. ERFs showed a prominent short-latency component at

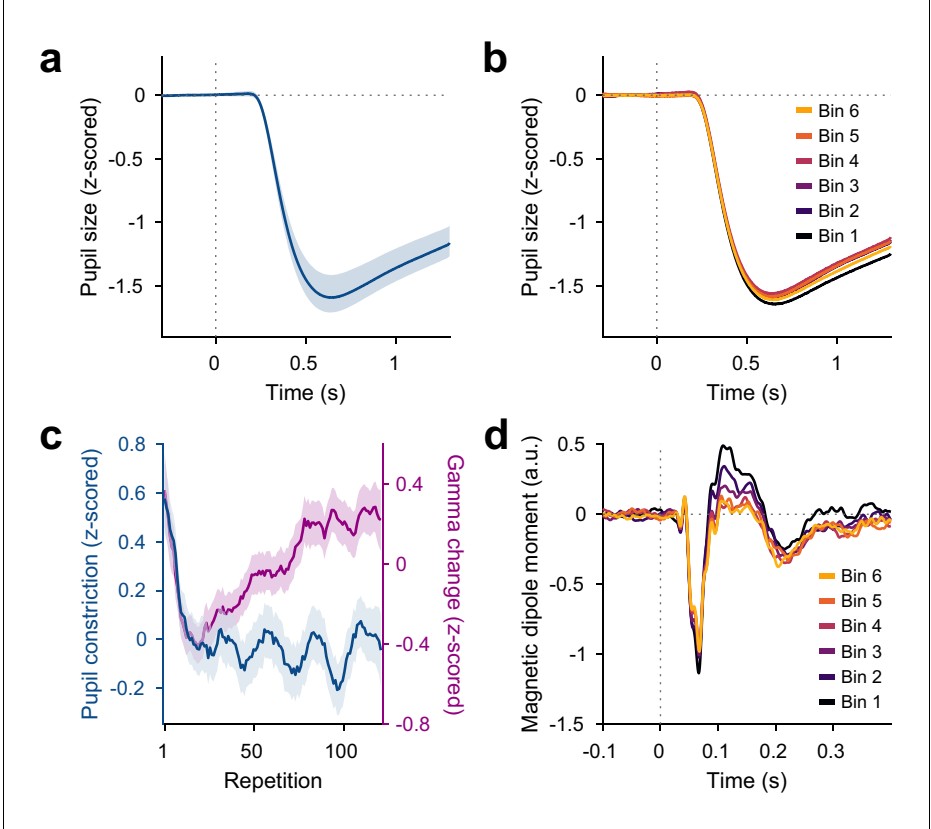

**Figure 4.** Repetition effects on pupil constriction and ERF. (**a**) Average pupil size as a function of time post-stimulus onset, z-scored relative to the baseline. A pupillary light reflex to the luminance increase at stimulus onset can be seen. All pupil plots exclude block A because pupil size at the beginning of the experiment was confounded by slow adaptation to the projector illumination (see *Figure 4—figure supplement 1a*). (**b**) Same as (**a**), but averaged for bins of 20 stimulus repetitions each. (**c**) Blue: Per-repetition average pupil constriction (defined as the per-trial difference between mean pupil size during the 300 ms baseline period and the 0.5–1.2 s post-stimulus period, z-scored within subjects). Violet: Per-repetition stimulus-induced gamma power change in V1/V2 (z-scored within subjects), for comparison. The average and the 95% bootstrap confidence intervals were computed using a five-trial-wide running window. (**d**) Magnetic dipole moment in V1/V2 in response to stimulus onset, averaged for bins of 20 stimulus repetitions each.

The online version of this article includes the following figure supplement(s) for figure 4:

**Figure supplement 1.** Per-trial changes in pupil constriction and ERF magnitude.

---

55–70 ms post-stimulus-onset, which we refer to as C1, and a longer-latency component at 90–180 ms post-stimulus-onset, which we refer to as C2. The per-trial magnitudes of both C1 and C2 decreased with stimulus repetition (*Figure 4d*, *Figure 4—figure supplement 1b,c*). Specifically, both C1 and C2 showed a stimulus-specific decrease in magnitude during the first 10 repetitions of a stimulus (C1: $p<5*10^{-5}$, C2: $p = 0.03$) and over further stimulus repetitions (C1: $p = 0.001$, C2: $p<2*10^{-16}$) above and beyond a stimulus-unspecific decrease in magnitude over trial numbers, which occurred only for C2 (C2: $p<2*10^{-9}$; for C1 the $CI_{95\%}$ included 0). As for gamma power, this repetition effect persisted over time: Both C1 and C2 showed a decreased magnitude during the repetition block A2, beyond the level predicted by all other factors (C1: $p = 0.001$, C2: $p = 0.015$).

Over the first 10 repetitions of a given stimulus, both ERF magnitude and gamma power showed decreases. Over the remaining repetitions of a given stimulus, ERF magnitude showed further decreases, while gamma power showed increases. While the changes in C1 component magnitude did not correlate across subjects with the changes in gamma power during early ($p = 0.30$) or late ($p = 0.19$) stimulus repetitions, the changes in C2 component magnitude did correlate across

subjects with the changes in gamma power during early ($r_{Spearman} = 0.38, p = 0.038$) and late ($r_{Spearman} = -0.55, p = 0.002$) stimulus repetitions.

## Granger causality in the gamma band increases with stimulus repetition, especially for feedforward connections

Previous studies in macaques and humans found that Granger causality (GC) between cortical areas in the gamma band is stronger in the anatomically defined feedforward direction, whereas GC in the alpha-beta band is stronger in the feedback direction (*Bastos et al., 2015*; *Michalareas et al., 2016*). We repeated the core analysis of *Michalareas et al., 2016* for the present dataset and found a similar pattern of results (*Figure 5a,b, Figure 5—figure supplement 1*).

The above-described effects of stimulus repetition on gamma might be accompanied by corresponding changes in Granger causality. The analysis of MEG source-level GC requires a sufficient number of data points. Therefore, we compared GC computed over trials 11–50 (i.e., excluding the first ten trials showing the early gamma-power decrease) with GC computed over trials 81–120. *Figure 5a,b* shows two example interareal GC spectra with repetition-related increases in feedforward gamma GC. Across all area pairs, significant GC changes with stimulus repetition were strongly clustered in the gamma band, while no significant changes in the alpha-beta band were found (*Figure 5c*). Correspondingly, we focused the following analyses on the gamma band.

The estimation of the GC metric can be affected by the signal-to-noise ratios (SNRs) of the respective sources. One conservative test of GC directionality time-reverses the involved signals, which leaves the SNRs unchanged, but reverses temporal relations (*Haufe et al., 2012*; *Vinck et al., 2015*). Therefore, GC directionality that switches upon time reversal is most likely not due to SNR differences. In the following, we report only repetition-related effects that were both significant before time reversal and significant, with opposite directionality, after time reversal (*Vinck et al., 2015*).

With stimulus repetition, across all between-area pairs, feedforward gamma GC increased from V1 to V2/V3/V4, and from V3/V3AB/V4 to several areas further up the dorsal and ventral streams (*Figure 5d,e*). Feedback GC onto areas V1–V4 also increased. Across all significant repetition-related GC changes, feedforward connections increased more strongly than feedback ones (*Figure 5f*, $p<0.001$). We considered whether the observed changes in gamma GC were purely driven by changes in gamma power. The respective gamma-power changes (calculated similarly to the GC changes) are shown as a colored vertical bar to the right of *Figure 5e*. As can be seen, gamma-power changes tended to decrease with hierarchical level. By contrast, gamma-GC changes were strongest for GC that originated from intermediate levels and was directed to high levels. Furthermore, gamma-GC changes remained significantly above zero when we regressed out gamma-power changes in both areas of the area pairs ($CI_{95\%} = [0.0005\,0.0010]$, $p<2*10^{-6}$). This demonstrates that the changes in gamma GC were not purely driven by changes in the signal-to-noise ratio of the gamma band.

## Low-frequency baseline power increases with time-on-task, independent of stimulation

As described before (*Benwell et al., 2019*), baseline power in the subject-specific alpha-band increased with trial number (*Figure 3—figure supplement 3a*, $p<2*10^{-16}$), independent of the stimulus.

## Discussion

In summary, repeated presentations of a visual stimulus induced gamma-band activity in early and intermediate visual areas that decreased over the initial 10 repetitions and subsequently increased over further repetitions. Crucially, when stimuli were switched, this pattern repeated. This strongly suggests that the changes in the neuronal circuits that underlie the observed gamma-power increase are specific to the repeated stimulus and do not equally affect the processing of other stimuli. Gamma peak frequency increased over repetitions and did not show distinct changes for the first few repetitions. The stimulus-specific increases in gamma power and frequency with repetitions showed a stimulus-specific memory effect, in the sense that some enhancement persisted over 25 minutes of

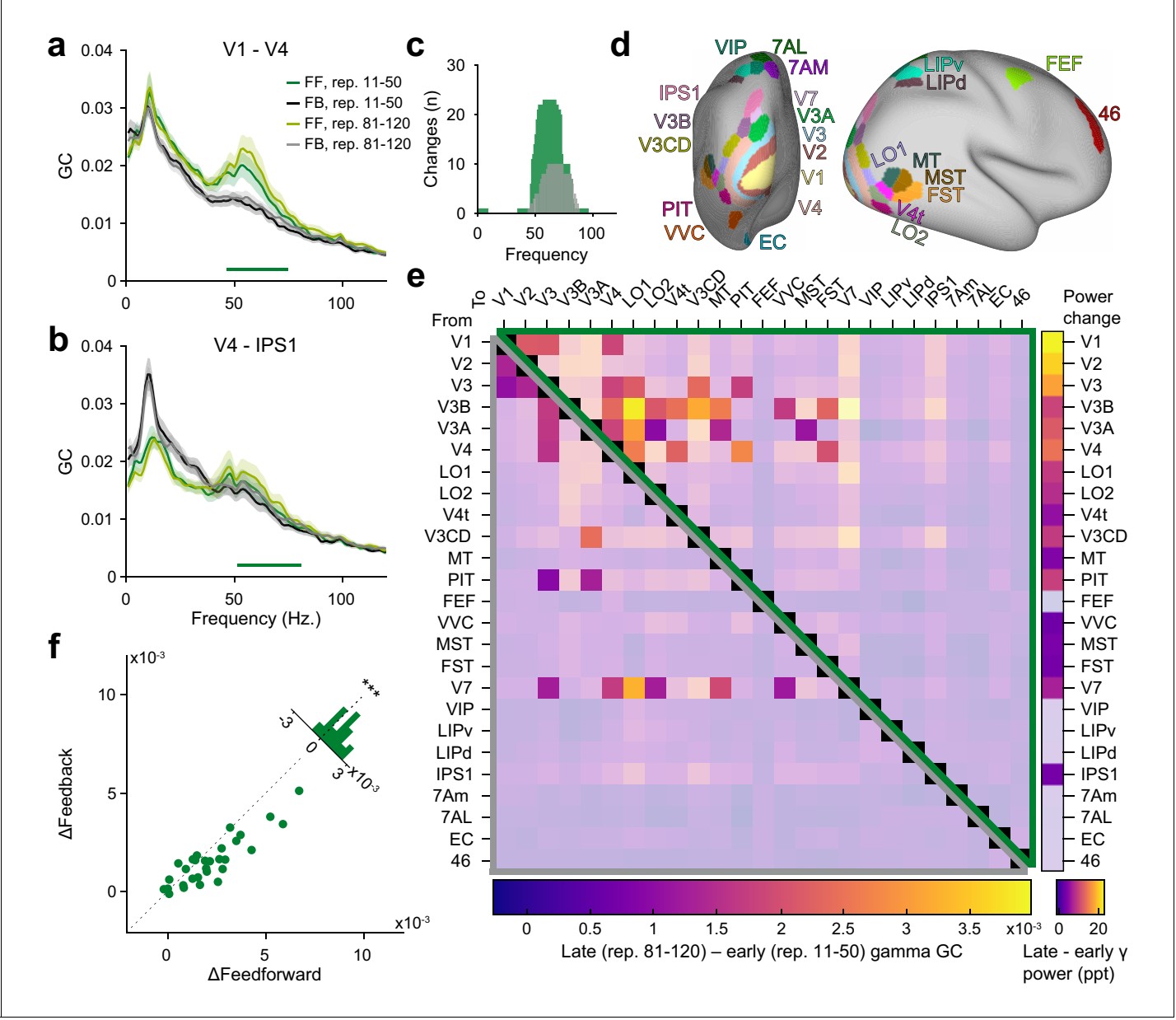

**Figure 5.** Repetition effects on GC are strongest for gamma in the feedforward direction. (**a**) Bivariate GC spectra between areas V1 and V4 (FF = feedforward, i.e. V1-to-V4, FB = feedback, i.e. V4-to-V1). GC was separately computed for early repetitions (trials 11–50, i.e. after the early gamma decrease) and late repetitions (trials 81–120). Error regions reflect 95% CIs. Inferential statistics are based on a non-parametric permutation test cluster-corrected for multiple comparisons across frequencies (*Maris and Oostenveld, 2007*). Horizontal green bar indicates significant cluster for FF GC. (**b**) Same analysis as in (**a**), but for areas V4 and IPS1, with feedforward being V4-to-IPS1 and feedback being IPS1-to-V4. (**c**) Total number of per-frequency significant differences between late and early repetition GC spectra between all areas (green = feedforward, gray = feedback). (**d**) All areas used for the analysis, plotted onto a semi-inflated average cortical surface. Area and surface definitions were taken from the HCP MMP1.0 atlas (*Glasser et al., 2016a*). (**e**) Changes in gamma GC from early to late trials, separately for the feedforward direction (upper matrix half, enclosed by green triangle) and the feedback direction (lower matrix half, enclosed by gray triangle). Non-significant matrix entries are gray masked. To be considered significant, matrix entries had to pass a $t_{max}$-corrected paired permutation test including time-reversal testing (*Vinck et al., 2015*). Inset right: Changes in gamma power for each brain area from early to late repetitions (significance based on a $t_{max}$-corrected paired permutation test; non-significant areas are gray masked). (**f**) The analysis of (**e**) was repeated per subject, and for the individually significant matrix entries, GC changes were averaged, separately for the feedforward (x-axis) and feedback (y-axis) direction; each dot corresponds to one subject. Across subjects, repetition-related GC changes were larger in the feedforward than the feedback direction ($p<0.001$).

The online version of this article includes the following figure supplement(s) for figure 5:

**Figure supplement 1.** Conceptual replication of *Michalareas et al., 2016*.

stimulation with different stimuli. This suggests that the repetition-driven network changes are at least partially persistent. Furthermore, gamma-band Granger causality increased with stimulus repetitions, especially from early visual areas in the anatomically defined feedforward direction. In addition, the magnitude of early ERF components decreased linearly with stimulus repetitions.

The repetition-related changes occurred over two different timescales, potentially indicative of two distinct but co-occurring processes. Over the first 10 stimulus repetitions, gamma power and pupil constriction decreased, and the slopes of their decreases were correlated across subjects. Over the remaining repetitions, gamma power increased continuously, while pupil constriction remained at the lower level. This pattern of changes is consistent with the superposition of an exponential decay, seen in gamma power and pupil constriction, with a slow and steady increase, seen in gamma power. In support of this scenario, two other parameters of neuronal activity showed such changes over all repetitions of a given stimulus: Gamma peak frequency steadily increased, and the magnitude of an early ERF component steadily decreased.

By extending research on gamma repetition enhancement from non-human primate local field potential recordings to human source-localized MEG, we could show remarkable similarities between gamma-band activities and their repetition-related changes, measurable in both species and recording techniques – see the companion paper (*Peter et al., 2020*). Notably, the existing studies with animals are limited to two to four subjects and thereby to an inference on those samples, whereas the present MEG study recorded from 30 subjects and thereby allowed an inference on the population.

Analyzing MEG recordings in source space suffers from uncertainties in spatial localization. Nevertheless, careful head stabilization and exclusion of participants with excessive head movements, as implemented in this study, enables a spatial resolution between 0.45 mm and 7 mm (*Nasiotis et al., 2017*). When analyzing Granger causality, it is important to stress that GC does not necessarily imply the existence of true neuronal interactions between time series, but merely implies predictability of one time series by another (*Kispersky et al., 2011*). Additionally, common noise and field spread in signals analyzed using GC can lead to spurious inferred connectivity, which can, however, be mostly alleviated using time-reversal testing, as used in this study (*Haufe et al., 2012*; *Vinck et al., 2015*).

## Strong neuronal responses to unexpected stimuli

In our recordings, the first trial of each block showed strong induced gamma power, followed by a decrease over the following nine trials. As subjects had not been informed about the different orientations, their blocked order, or the block length, stimulus switches were unexpected. Furthermore, as grating stimuli were not shown during training and subjects were recruited from the general public, grating stimuli were mostly novel. Unexpected and novel stimuli have been shown to induce stronger neuronal responses in early visual cortex: In an fMRI paradigm, subjects showed hemodynamic response increases in V1 when a presented grating had a different orientation to the one expected by the subjects (*Kok et al., 2016*). In an MEG study, in which subjects learned that presented visual stimuli followed a specific stimulus sequence, the occipital cortex showed stronger activation when the expected stimulus sequence was violated or when stimuli were presented that the subjects were not familiar with (*Manahova et al., 2018*).

Unexpected stimuli, or transient increases in task difficulty created by them, likely engage mechanisms of attention and/or arousal, which can be gauged by measuring pupil size. Pupil diameter has been linked to arousal in several studies (*de Gee et al., 2017*; *Peinkhofer et al., 2019*). Pupil dilation can best be used in studies that avoid changes in stimulus luminance. Paradigms including luminance increases, as used here, induce pupil constrictions, referred to as the pupillary light reflex, which can also be influenced by arousal, attention, and stimulus novelty (*Binda et al., 2013*; *Naber et al., 2013*). In our data, pupil constrictions were strong on initial stimulus presentation, decreasing over the first ten repetitions and remaining low for further repetitions. Gamma power showed correlated dynamics for the initial 10 presentations of a given stimulus, but showed increases for further repetitions. This is consistent with a scenario in which stimulus novelty leads to strong gamma and pupil responses for the initial presentation of a stimulus and the rapid decline thereafter, and other mechanisms lead to the steady increase in gamma for later repetitions.

In the present study, the late increases brought gamma power approximately back to the level of the first few repetitions in a block. This initial level might therefore be interpreted as the maximal possible level, which is lost during early repetitions and slowly regained during later repetitions.

However, data obtained with invasive recordings in macaque monkeys show that the gamma-power decrease during early trials can be strongly exceeded by the increase during later trials (*Brunet et al., 2014*; *Peter et al., 2020*).

### Firing rate repetition effects in early visual cortex

Firing rates in early visual cortex decrease with both stimulus repetitions over neighboring trials and stimulus familiarization over days to months. In macaque areas V1 and V4, the across-trial repetition of stimuli induced strong firing rate decreases over the first few repetitions as well as smaller firing rate decreases over further repetitions (*Brunet et al., 2014*; *Peter et al., 2020*). In macaque V4, firing rate decreases have also been reported within single trials, between the first and immediately following second presentation of a given stimulus (*Wang et al., 2011*). In mouse V1, such a within-session repetition-driven decrease in neuronal activity, measured using calcium imaging, occurred as a sparsification of the neuronal response: While most measured neurons decreased their activity with repetitions, a small set of strongly driven neurons stayed continually active even after repetitions (*Homann et al., 2017*).

Similar effects have also been found when animals were familiarized with a set of stimuli over multiple days and were then shown both the stimuli they were familiarized with, as well as novel stimuli. In macaque V2, firing rate responses were smaller for familiar than for novel images from 100 ms post-stimulus onset (*Huang et al., 2018*). Such decreases in neuronal responses with familiarity have also been linked to response sparsification: When macaques were trained to identify grating orientations over several months, tuning curves of V1 neurons responsive to orientations close to the trained orientation steepened at the trained orientation (*Schoups et al., 2001*). In large populations of neurons recorded in macaque IT, putative excitatory neurons showed higher selectivity to images the monkeys had been familiarized with over months compared to novel images (*Lim et al., 2015*; *Woloszyn and Sheinberg, 2012*).

### Gamma repetition effects in early visual cortex of primates

Studies measuring gamma-band responses in early visual cortex over stimulus repetitions generally reported gamma power decreases over a single repetition or prolonged exposure paradigms, and gamma power and frequency increases over higher repetition numbers. In anesthetized macaque V1, when the neuronal response to an oriented grating stimulus was adapted by presentation for 40 s (plus additional top-up presentations), a subsequent display of the same orientation induced weaker gamma power, whereas other orientations induced stronger gamma power (*Jia et al., 2011*). In addition, decreases in broadband gamma power but increases in broadband gamma spike-field locking with one-shot adaptation have also been recorded in awake macaque V4, and have been hypothesized to be driven by synaptic depression (*Wang et al., 2011*). In human MEG and fMRI, the second presentation of familiar visual stimuli induced weaker gamma-band power and weaker hemodynamic responses in early visual areas than the first one (*Friese et al., 2012a*; *Friese et al., 2012b*). A study of macaque V1 and V4 activity (*Brunet et al., 2014*), using up to 600 repetitions of few similar grating stimuli, found that LFP gamma power and frequency in both areas, and their coherence, increased with the logarithm of repetition number. Furthermore, stimulus repetition also affected gamma spike-field locking in V4: For putative interneurons, it increased, and for putative pyramidal cells, there was a positive relationship between their stimulus drivenness and the slope of repetition-related changes in locking. The companion paper to the one presented here investigated repetition-related gamma increases in macaque V1 and found that they are also specific to the repeated stimulus, have some persistence, and generalize to natural stimuli (*Peter et al., 2020*).

### Repetition-related increases in the characteristic rhythms of other organisms

Changes in LFP power with stimulus repetition have also been reported in other organisms: In the locust antennal lobe, odor repetition decreased firing of excitatory neurons to a limited set of reliably firing neurons and increased power and spike-field locking in the dominant odor-driven LFP oscillation (the beta band) in a stimulus-specific fashion (*Bazhenov et al., 2005*; *Stopfer and Laurent, 1999*). In the rat, odor-driven gamma-band oscillations in the olfactory bulb and the

orbitofrontal cortex also increased with odor repetition during task learning (*Beshel et al., 2007*; *van Wingerden et al., 2010*).

## Potential mechanism of late gamma increase as local circuit learning

Oscillatory neuronal activity can interact with Hebbian spike-timing-dependent plasticity (STDP). This can, for example, lead to changes in synaptic weights between excitatory neurons (E-E) that enhance their temporal synchronization and establish excitatory cell assemblies (*Arthur et al., 2005*; *Cassenaer and Laurent, 2007*; *Suri and Sejnowski, 2002*) as well as shorten oscillatory cycles (*Börgers, 2017*). However, changes in E-E synaptic weights would not explain the observed decreases in firing rates and ERFs and increases in inhibitory gamma locking (reported here; *Brunet et al., 2014*; *Peter et al., 2020*). We would like to speculate on a possible neuronal mechanism consistent with these findings as well as the reported increases in gamma power, frequency, and interareal gamma coherence (*Figure 6*).

When visual stimulation induces gamma-band activity in awake primate V1 (*Brunet et al., 2015*; *Jia et al., 2011*; *Kreiter and Singer, 1992*; *Uran et al., 2020*), the resulting gamma cycles contain systematic sequences: The better a neuron is driven by a given stimulus, the earlier it spikes in the gamma cycle (*Fries et al., 2007*; *Havenith et al., 2011*; *König et al., 1995*; *Vinck et al., 2010*). This is likely due to the fact that the gamma cycle contains a characteristic sequence of excitation and inhibition (*Atallah and Scanziani, 2009*; *Csicsvari et al., 2003*; *Hasenstaub et al., 2005*; *Vinck et al., 2013*). Excitation triggers inhibition, and when inhibition decays, the most driven neurons are the first to overcome inhibition and spike. Their spiking leads to a new rise in inhibition, and only sufficiently driven neurons spike before the rising inhibition prevents the least driven neurons from spiking at all (*de Almeida et al., 2009*). Thus, on average, the most driven excitatory neurons ($E_{strong}$) spike first, followed by spiking of local inhibitory neurons ($I_{local}$), while less driven excitatory neurons ($E_{weak}$) spike during and after the inhibitory neurons, if at all. This sets up an $E_{strong}$-$I_{local}$-$E_{weak}$ spiking sequence.

If two neurons spike for some time with a systematic temporal relationship, this can lead to changes in their mutual synaptic inputs, a phenomenon referred to as spike timing-dependent plasticity (STDP) (*Caporale and Dan, 2008*; *Hennequin et al., 2017*). The precise pattern of synaptic strengthening and weakening as a function of the relative spike timing varies across neuron types and brain areas (*Hennequin et al., 2017*). One well-established pattern is referred to as Hebbian STDP: Synapses from the leading neuron spiking few milliseconds before the lagging neuron are strengthened, whereas synapses in the other direction are weakened. This pattern has, e.g., been described for synapses of excitatory neurons onto inhibitory neurons in rat visual cortex (*Huang et al., 2013*). This Hebbian STDP, together with the abovementioned $E_{strong}$-$I_{local}$-$E_{weak}$ sequence during gamma cycles, would lead to a strengthening of the synapses from $E_{strong}$ to $I_{local}$, and to a weakening of the synapses from $E_{weak}$ to $I_{local}$. Note that the timescales of spike sequences in the gamma cycle and of spike relationships leading to STDP are in reasonably good agreement. For synapses of inhibitory neurons onto excitatory neurons, the described STDP patterns are overall more diverse. Yet, a Hebbian-type I-to-E STDP has been found in rat entorhinal cortex (*Haas et al., 2006*). Together with the gamma-related $E_{strong}$-$I_{local}$-$E_{weak}$ sequence, this could lead to strengthening of synapses from $I_{local}$ to $E_{weak}$ and weakening of synapses from $I_{local}$ to $E_{strong}$.

Through this interplay between the gamma cycle and STDP, the activation of $E_{strong}$ neurons during the repeated presentation of a given stimulus would increase the impact of $E_{strong}$ onto $I_{local}$ neurons. $E_{strong}$ spiking would trigger $I_{local}$ spiking with more efficiency and shorter latency, leading to stronger and earlier $I_{local}$ spiking, and thereby more gamma-locked $I_{local}$ spiking. This could explain the observed shorter gamma cycles (i.e., higher gamma frequency) and overall stronger gamma power measured here and in macaques (*Brunet et al., 2014*; *Peter et al., 2020*), and the increasing gamma locking of inhibitory neurons (*Brunet et al., 2014*). At the same time, these stronger and more synchronized bouts of $I_{local}$ spiking would enhance the impact of $I_{local}$ neurons onto $E_{weak}$ neurons. Additionally, the inhibition of $E_{weak}$ neurons would be further enhanced by the strengthened $I_{local}$-to-$E_{weak}$ synapses. The strong bouts of $I_{local}$ spiking might in principle also enhance the $I_{local}$ feedback inhibition onto $E_{strong}$ neurons. However, this effect might be balanced by the weakened $I_{local}$-to-$E_{strong}$ synapses. In sum, this could lead to maintained firing of $E_{strong}$ together with reduced firing of $E_{weak}$ neurons, and thereby explain overall reduced firing rates and implement a winner-

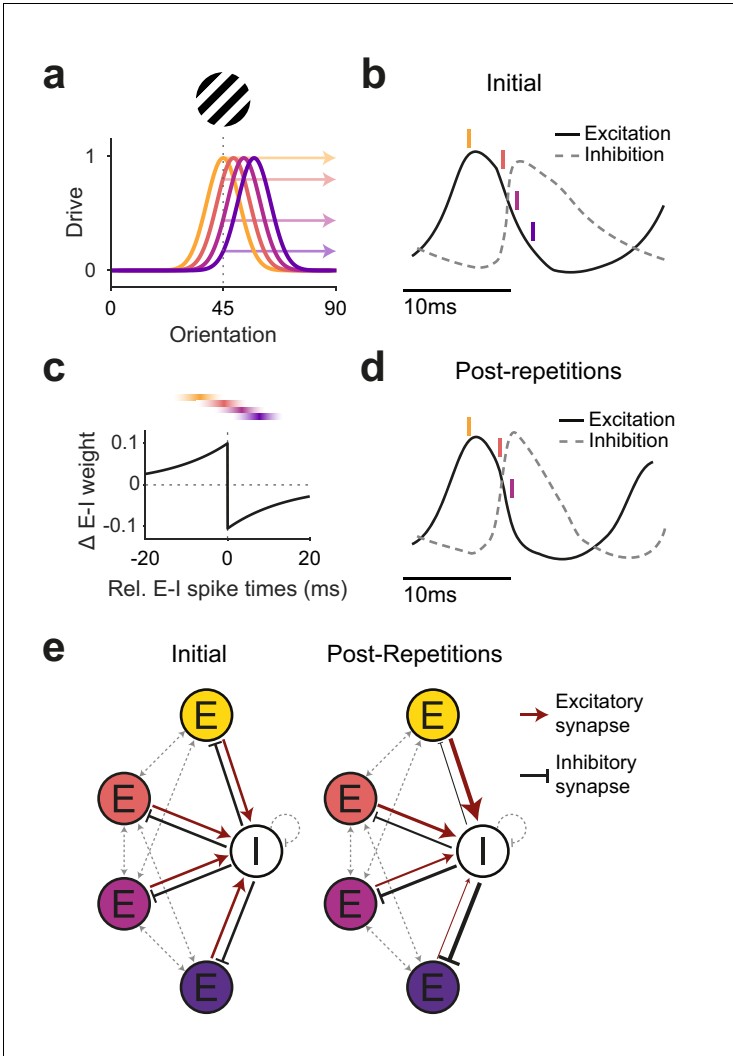

**Figure 6.** Illustration of a potential neuronal mechanism of repetition-induced gamma changes. (a) Tuning curves of four example excitatory neurons (colored lines) and the drive they receive (colored arrows) for a stimulus of a given orientation (shown above the panel). (b) Local average excitatory inputs (black solid curve) and inhibitory inputs (gray dashed curve), adapted from Figure 6D of *Salkoff et al., 2015*, during a gamma cycle. Inhibitory inputs systematically lag excitatory inputs by a few milliseconds. Colored vertical lines indicate mean spike times of the four example neurons, color-coded according to (a). Their spike latencies during the gamma cycle are determined by their stimulus drive (*Vinck et al., 2010*). (c) A Hebbian STDP kernel, aligned to the average time of inhibitory spiking. The relative E-I spike timing between strongly driven/weakly driven excitatory neurons and the inhibitory neuron pool induces E-to-I synaptic weight increases/decreases, respectively. Note that the spike times shown in (b) are illustrations of the mean spike times of the respective neurons during the gamma cycle, whereas experimentally observed spike time distributions show substantial cycle-by-cycle variability. Thereby, for the two neurons with the strongest (yellow) and weakest (blue) drive, spike times occur almost exclusively during the positive or negative part of the STDP kernel, respectively. For the neuron with the second-strongest drive (red), spike times mostly overlap with the positive part, yet also partly with the negative part of the STDP kernel, and the reverse holds for the neuron with the second-weakest drive (purple). (d) The proposed mechanism should result in a modified E-I dynamic: Strengthened synaptic weights from strongly driven excitatory neurons to the inhibitory neuron pool accelerate the excitation-driven inhibition, thereby shortening the gamma cycle and increasing MUA-LFP gamma locking. (e) The proposed mechanism strengthens synaptic weights from strongly driven excitatory neurons to the local inhibitory interneuron pool. Furthermore, it strengthens inhibitory synaptic weights from the local inhibitory interneuron pool to the more weakly driven excitatory neurons.

take-all mechanism that sharpens the population firing rate representation (*de Almeida et al., 2009*; *Homann et al., 2017*; *Lim et al., 2015*).

Beyond these local effects, the overall increase in gamma strength and the stronger focusing of $E_{strong}$ spiking during the early gamma cycle would likely enhance the impact of the local $E_{strong}$ neurons onto their postsynaptic target neurons in other areas (*Salinas and Sejnowski, 2000*). This is consistent with the observed repetition-related enhancement of V1–V4 gamma coherence (*Brunet et al., 2014*) and of feedforward gamma GC (this study). Thereby, the sharpened neuronal population response might be communicated more efficiently (*Lewis et al., 2021*). Increased synchronization would compensate for overall lower firing rates, thereby allowing the visual system to keep or improve behavioral performance with less neuronal activity (*Gotts et al., 2012*). Such changes should be specific to the activated cell assembly, extend over time and be robust to deadaptation, as shown in this study.

## Materials and methods

### Participants
Participants were recruited from the general public until 30 had successfully completed the experiment. Twenty of the 30 participants were male. As they were recruited via general job advertisements, most of them had not participated in other neuroscientific experiments before. They were of an average age of 22 years (range: 19–28 years), had normal or corrected-to-normal vision, were free of metal implants, did not use medication during the study period except for contraceptives, and had never been diagnosed with any neurological or psychiatric disorder. All participants gave written informed consent. The study was approved by the ethics committee of the medical faculty of the Goethe University Frankfurt (Resolution E 36/18).

### Paradigm
Subjects were positioned in a dimly lit magnetically shielded room and undertook a simple change detection task. Visual stimuli were back-projected onto a screen 53 cm away from their eyes using a Propixx projector (Resolution: 960*520 px, 1440 Hz refresh rate). Eye position and pupil size were measured with an infrared eye tracker (EyeLink 1000). Once the subject fixated a central fixation spot for 0.45 s, the trial was initiated. It consisted of a 1 s baseline interval with a gray screen, 0.3–2.0 s (randomized, Cauchy-distributed with $x_0 = 1.65s, FWHM = 0.2s$) of visual stimulation, followed by a to-be-detected change. The stimulus was a centrally presented square wave grating with anti-aliasing (i.e., slightly rounded edges), with a diameter of 22.9 degrees of visual angle (dva), a spatial frequency of 4 cycles/dva, and one of the following orientations: 22.5 deg, 67.5 deg, 121.5 deg, 175.5 deg. Stimulus luminance averaged over the grating patch, measured with a Minolta CS-100A, was 1300 cd/m$^2$. The change was a contrast reduction of the entire grating by 50%, which served as a cue to report the simultaneously applied rotation of the grating to the left or the right by 0.25–0.9 deg. The rationale for the combination of a salient contrast change with a threshold-level rotation was the following: The contrast change was perceived on each trial and cued the subjects to report the rotation, yet the rotation was titrated to maximize the sensitivity for detecting accuracy changes. Five percent of trials were change-free catch trials. Subjects were instructed to speedily report the direction of the orientation change, using a button press with their index (for left rotations) or middle (for right rotations) finger. Presses were followed by the 0.5 s presentation of a smiley, which served as positive feedback irrespective of accuracy. This was followed by the presentation of the fixation point for the next trial, which was self-initiated within 0.5–4 s, when the subject attained fixation.

For each subject, a total of 600 trials were recorded, composed of 5 blocks of 120 trials. Blocks were labeled A, B, C, D, A2, with the letters randomly assigned (per participant) to one of the four orientations, and A2 constituting a repetition of block A. Note that trials proceeded seamlessly across block boundaries, i.e. there was no break, change or instruction of any kind between blocks, and subjects were instructed to disregard stimulus orientation. The whole experiment lasted 45 min on average, giving a time interval of approximately 27 min between the end of block A and the beginning of block A2. Before the experiment, subjects were trained on the task using white-noise disks instead of the grating stimulus.

## MEG recording

Data were recorded using an MEG system (CTF Systems) comprising 275 axial gradiometers, low-pass filtered (300 Hz), and digitized (1200 Hz). Subjects were trained before the experiment to avoid eye blinks during the baseline and stimulation period and to instead blink during each inter-trial interval. Therefore, blinks occurred in merely 3.9% of all analyzed trials. Subjects were positioned to minimize the distance between the occipital pole and the dewar helmet, head movement was minimized using memory foam cushions and a chin rest. Head position was continuously monitored throughout the experiment. Head drift >5 mm away from the initial head position was considered excessive. Excessive head drift, falling asleep, hardware malfunctions, or similar problems resulted in immediate abortion of the recording session and exclusion of the respective subject from the study. Any break or interruption to fix those problems would have interfered with the repetition protocol. In total, this exclusion applied to nine subjects, which were not counted toward the 30 subjects reported here.

## Data analysis

Data were analyzed using custom Matlab, R, and Python code and the Fieldtrip (*Oostenveld et al., 2011*) and Freesurfer (*Fischl, 2012*) toolboxes. Line noise was removed using discrete Fourier transform filters. Data were cut into epochs from 1 s before stimulus onset up to the stimulus change. Trials with stimulus changes before 1.3 s after stimulus onset, trials with missing/early responses, and catch trials were removed. Data segments containing SQUID jumps, muscle artifacts, and blinks were labeled as artifacts. Artifact-free parts of the respective epochs were used for the analyses described below if they contained data for the full respective analysis window lengths, which was the case for 76% of trials. Data from the repetition block A2 were only part of analyses investigating effects of the repetition block. Microsaccades were detected using the algorithm described by *Engbert et al., 2002*.

Subject-specific theta, alpha, beta, and gamma peak frequencies were determined using 1/f-removal (by fitting and subtracting a linear fit to the semilog power spectrum) and subsequent fitting of Gaussians to the per-participant average stimulus-induced power spectra at the driven V1/V2 dipole (*Haller et al., 2018*). This procedure found a subject-specific gamma peak for all subjects. If no clear subject-specific theta/alpha/beta peak could be found, a representative peak frequency of the other subjects (theta: 6 Hz, alpha: 10 Hz, beta: 20 Hz) was taken instead.

## Source localization

Analyses at the subject-specific theta-, alpha-, beta-, and gamma-band peaks used source projection by means of dynamic imaging of coherent sources (DICS) beamformers (*Gross et al., 2001*). All other analyses used source projection by means of linearly constrained minimum variance (LCMV) beamformers (*Van Veen et al., 1997*) and were run in source space, except for *Figure 3—figure supplement 1e,f*, which explicitly shows sensor level results. Both, the DICS and the LCMV beamformers, were computed without regularizing the covariance matrix ($\lambda = 0\%$) and estimated spatial filters for all vertices of both hemispheres of the 32 k HCP-MMP1.0 atlas (*Benson et al., 2018*; *Glasser et al., 2016a*). This atlas was registered to subject-specific MRIs (T1: MPRAGE, 1 mm$^3$) using Freesurfer and the Connectome Workbench (*Glasser et al., 2016b*). Area-specific analyses averaged their results (power, change coefficients, granger coefficients) over LCMV dipoles using the 180 parcels of this atlas. Event-related fields and time–frequency plots were computed for the participant-specific LCMV dipole showing the strongest stimulus-induced gamma power response as a functional localizer for visual areas, which fell into V1 or V2. We restricted between-area Granger causality analyses to areas within an MEG-based visual hierarchy (*Michalareas et al., 2016*).

## Spectral and ERF analyses

Power over all frequencies was computed for 1 s baseline (−1 s to stimulus onset, power averaged within blocks) and stimulus (0.3–1.3 s post-stimulus onset) data periods. These periods were cut into 50% overlapping windows. For the analysis of frequencies up to 20 Hz, we used 500 ms window length resulting in a spectral resolution of 2 Hz; for frequencies above 20 Hz, we used 333 ms windows, resulting in 3 Hz resolution. The windowed data were demeaned and detrended, then zero-padded to 1 s. For frequencies below 20 Hz, they were Hann tapered, for frequencies above 20 Hz,

they were Slepian-window multitapered using three tapers to achieve ±3 Hz smoothing. Finally, the data were Fourier transformed.

For the time-frequency plot in *Figure 2d*, the same analysis was run, except with zero-padding to 4 s and with windows centered on each time point (at 1200 Hz temporal resolution) between −1 s and 1.3 s relative to stimulus onset.

To quantify power changes at subject-specific theta, alpha, beta, and gamma peak frequencies, we used the power-change value at the individual peak frequency bin of the respective band (see Data analysis; *Haller et al., 2018*).

On the source level, we fit per-repetition band power with two regression lines: one line from trial 1–10, and a separate line from trial 11–120. To find this breakpoint between early gamma power decrease and late gamma power increase, we averaged per-repetition gamma power over the four non-repeated blocks, z-scored within each participant and averaged over participants. The resulting subject-average time course over repetitions was smoothed with a moving average over 10% of all repetitions and a broken line fit was applied. The best-fitting breakpoint was found to be repetition 10.

The approach used to estimate trial-averaged theta/alpha/beta/gamma peak frequencies (see Data analysis) could not be used on a per-trial basis because individual trial spectra were too noisy to estimate the full number of parameters. Therefore, gamma-peak frequency per participant and per trial was estimated as follows: The spectra of per-trial power change relative to baseline were fit with a Gaussian function, and if the fit Gaussian was of positive amplitude and showed a location term between 2 and 120 Hz, the location term was taken as the per-trial gamma peak frequency.

To compute event-related fields (ERFs), source-localized time courses from −0.2 s to 0.6 s relative to stimulus onset were low-pass filtered using an acausal Gaussian filter kernel (−6 dB at 80 Hz), baselined, and averaged.

## Granger spectral analyses

As MEG source-localized Granger causality is too noisy to be determined on a single-trial basis, we pooled trials 11–50 of blocks A-D as early repetitions and trials 81–120 of blocks A–D as late repetitions. Trials 1–10 were not included, as they contained the sharply decreasing gamma power at the beginning of each block.

To determine between-area Granger causality, sensor-level data from 0.4 s to 2 s post-stimulus onset were cut into 50% overlapping 500 ms windows. Each window was detrended by subtracting a Hann-taper-weighted regression fit, and subsequently Hann-tapered, zero-padded to 1 s length, and Fourier transformed. The resulting complex Fourier spectra were multiplied with the LCMV filters to transform them into source space, where we used them to define between-dipole cross-spectral densities (CSDs). Bivariate granger spectra between dipoles were then computed using non-parametric spectral matrix factorization of the CSD matrices (*Dhamala et al., 2008*) and averaged over all dipoles belonging to an atlas parcel pair.

We tested for differences between early and late GC spectra using cluster-based nonparametric significance testing over frequencies (*Maris and Oostenveld, 2007*). This was done separately for each between-area pair and for each direction (feedforward and feedback). To define area-pair connections as feedforward or feedback, we referred to an MEG-based definition of the human visual hierarchy (*Michalareas et al., 2016*).

We analyzed which area pairs showed changes in GC values between early and late trials: We compared GC values between early and late trials, across subjects, using a non-parametric permutation test with $t_{max}$-based correction for the multiple comparisons across area pairs. To test whether any results could be due to changes in signal-to-noise ratio or to between-area power differences, we performed two control analyses: (1) We repeated this analysis after time-reversing the sensor-level data. We only report effects that were significant in both time directions and flipped their change directionality with time-reversal (*Haufe et al., 2012*; *Vinck et al., 2015*). (2) We fit a regression of the gamma-power changes in both areas of each area pair onto the area pair changes in GC and tested the residuals against zero using a t-test and a bootstrapped confidence interval.

## Pupil size analyses

As mentioned above, data segments containing blink artifacts were excluded from the analysis. Pupil size data were then z-scored within each subject, the average over the last 300 ms before stimulus onset was subtracted per trial, and outlier values were identified as values more than 1.5 MAD (median average deviation) away from a 250 ms running median and replaced by linear interpolation. When pupil sizes of both eyes could be recorded in a subject, they were averaged before further analysis. As pre-stimulus pupil size was strongly decreasing at the beginning of the experiment (*Figure 3—figure supplement 1c* and *Figure 4—figure supplement 1a*), possibly due to effects on tonic pupil size of starting an experimental task (*Knapen et al., 2016*) or due to slow adaptation to the bright light of the projector, pupil size data from the first block and its repetition block were removed from all pupil size analyses. Pupil constriction was defined as the difference between mean pupil size during the 300 ms per-trial baseline and mean pupil size from 0.5 s to 1.2 s post-stimulus.

## Statistical analysis

Alpha was set to $\alpha = 0.05$, all reported tests were two-tailed, multiple comparison control was implemented using $t_{max}$ correction (*Blair et al., 1994*) unless otherwise noted. For all plots showing quantities developing over trial numbers, the mean and 95% bootstrap confidence interval lines were computed using a five-trial-wide running average. All plots show averages over subjects, no single-subject examples are shown.

We performed several hierarchical linear regression analyses, in which the dependent variable was either per-trial gamma power, ERF magnitude, or other per-trial measures, in which the independent variables were repetition number, overall trial number, the membership of a trial in the repetition block (categorical variable), pre-trial intertrial interval length, microsaccade rate, pupil constriction, the membership of a trial in the first 10 trials of a block (categorical variable), and in which random intercepts were fit for subject identity and stimulus orientation:

$$y_{trial,subject} = \beta_0 + Subject_0 + Orientation_0 + \beta_1 repetition\,number_{trial} + \beta_2 trialnumber_{trial} + \\ \beta_3 repetition\,block_{trial} + \beta_4 ITI_{trial} + \beta_5 microsaccade\,rate_{trial} + \beta_6 pupil\,constriction_{trial} + \\ \beta_7 early\,repetition_{trial}$$

We were interested in the effect of repetition number and included the other parameters as covariates. This model was separately fitted to the per-trial stimulus-induced gamma power, the per-trial ERF component magnitudes (C1 and C2, see Results text for definition), as well as to other reported outcomes of interest using the restricted maximum likelihood approach implemented in lme4 (*Bates et al., 2015*). Where necessary, this model was adapted: When setting one of the covariates as the dependent variable (as done for pupil constriction and microsaccade rate), it was removed from the independent variables. For pupil constriction, the repetition block parameter was removed, as the first block was also removed from the pupil size data (see above). Reported p-values were computed using Satterthwaite's approximation for degrees of freedom. Parameter confidence intervals were estimated using bootstrapping. Because the Satterthwaite approximation can be anticonservative (*Luke, 2017*), we only considered an effect as significant (and reported its p-value) if both the Satterthwaite-based p-values were significant and the bootstrap-based 95% confidence intervals did not include zero.

We investigated whether changes in gamma power were correlated across subjects to changes in other parameters, namely ERF size, pupil constriction, reaction time, and accuracy. Per subject, we fitted linear regressions using the same independent variables (except subject and orientation) as listed for the above linear-mixed model and using as dependent variable either gamma power, ERF size, pupil constriction, reaction times, or accuracy. Subsequently, the regression coefficients for the independent variable repetition number were correlated (Spearman's rank correlation) between gamma power and the other parameters. Because gamma power decreased across the first ten stimulus repetitions and increased across later trials, this was done separately for trials 1–10 and trials 11–120 (see Spectral and ERF analyses).

To estimate the time constants τ of early gamma power and pupil constriction decreases, we averaged per-repetition gamma power and pupil constriction over the four non-repeated blocks, z-scored within each subject and averaged over subjects. The resulting subject-average time courses over repetitions were fit as the sum of an exponential and a linear process over

repetitions, $y = ae^{\frac{x}{\tau}} + bx + c$. Resulting fit values were $R^2_{adj} = 0.68$ for gamma power and $R^2_{adj} = 0.36$ for pupil constriction.

## Acknowledgements

We thank Stan van Pelt for providing a dataset used in preliminary analyses and Julien Vezoli, Craig Richter, Georgios Spyropoulos, and Jarrod Dowdall for advice on data analysis. We also thank Gilles Laurent and David Poeppel for inspiration and advice on analysis and interpretation of the effects presented here. PF acknowledges grant support by DFG (SPP 1665 FR2557/1-1, FOR 1847 FR2557/2-1, FR2557/5-1-CORNET, FR2557/6-1-NeuroTMR, FR2557/7-1 DualStreams), EU (FP7-604102-HBP, FP7-600730-Magnetrodes), a European Young Investigator Award, NIH (1U54MH091657-WU-Minn-Consortium-HCP), and LOEWE (NeFF).

## Additional information

### Competing interests

Pascal Fries: has a patent on thin-film electrodes (US20170181707A1) and is beneficiary of a respective license contract on thin-film electrodes with Blackrock Microsystems LLC (Salt Lake City, UT),). P. F. is member of the Scientific Technical Advisory Board of CorTec GmbH (Freiburg, Germany), and managing director of Brain Science GmbH (Frankfurt am Main, Germany). P.F. declares no further competing interests. The other authors declare that no competing interests exist.

### Funding

| Funder | Grant reference number | Author |
| --- | --- | --- |
| Deutsche Forschungsgemeinschaft | SPP 1665 FR2557/1-1 | Pascal Fries |
| Deutsche Forschungsgemeinschaft | FOR 1847 FR2557/2-1 | Pascal Fries |
| Deutsche Forschungsgemeinschaft | FR2557/5-1-CORNET | Pascal Fries |
| Deutsche Forschungsgemeinschaft | FR2557/6-1-NeuroTMR | Pascal Fries |
| Deutsche Forschungsgemeinschaft | FR2557/7-1 DualStreams | Pascal Fries |
| European Union | FP7-604102-HBP | Pascal Fries |
| European Union | FP7-600730-Magnetrodes | Pascal Fries |
| European Union | European Young Investigator Award | Pascal Fries |
| NIH | 1U54MH091657-WU-Minn-Consortium-HCP | Pascal Fries |
| LOEWE | NeFF | Pascal Fries |

The funders had no role in study design, data collection and interpretation, or the decision to submit the work for publication.

### Author contributions

Benjamin J Stauch, Conceptualization, Software, Formal analysis, Investigation, Methodology, Writing - original draft, Writing - review and editing; Alina Peter, Conceptualization, Formal analysis, Writing - review and editing; Heike Schuler, Investigation, Methodology; Pascal Fries, Conceptualization, Formal analysis, Supervision, Funding acquisition, Methodology, Writing - original draft, Writing - review and editing

## Author ORCIDs
Benjamin J Stauch ⓘ https://orcid.org/0000-0002-4484-813X
Alina Peter ⓘ https://orcid.org/0000-0001-8497-6235
Pascal Fries ⓘ https://orcid.org/0000-0002-4270-1468

## Ethics
Human subjects: All participants gave written informed consent. The study was approved by the ethics committee of the medical faculty of the Goethe University Frankfurt (Resolution E 36/18).

## Decision letter and Author response
Decision letter https://doi.org/10.7554/eLife.68240.sa1
Author response https://doi.org/10.7554/eLife.68240.sa2

# Additional files

## Supplementary files
• Transparent reporting form

## Data availability
Per-trial data and code for statistical analyses have been uploaded and are available at https://doi.org/10.5281/zenodo.4269714. Preprocessing code has been uploaded as source code file 1.

The following dataset was generated:

| Author(s) | Year | Dataset title | Dataset URL | Database and Identifier |
|---|---|---|---|---|
| Benjamin JS, Alina P, Heike S, Pascal F | 2020 | Dataset for Stimulus-specific plasticity in human visual gamma-band activity and functional connectivity | https://doi.org/10.5281/zenodo.4269714 | Zenodo, 10.5281/zenodo.4269714 |

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
