## [Decision Letter]

**Acceptance summary:**

This combined MEG pupillometry study investigated stimulus-specific plasticity in human visual γ-band activity. The work was conducted thoroughly, and exhibits a high degree of technical proficiency. The results show that both gamma-band MEG and pupil size responses to visual stimuli adapt across stimulus repetitions. The claims are fully supported by the data and this work will be of broad interest to readers in the fields of non-human primate and human electrophysiology.

**Decision letter after peer review:**

Thank you for submitting your article "Stimulus-specific plasticity in human visual gamma-band activity and functional connectivity" for consideration by *eLife*. Your article has been reviewed by 2 peer reviewers, and the evaluation has been overseen by a Reviewing Editor and Chris Baker as the Senior Editor. The following individual involved in review of your submission has agreed to reveal their identity: Tomas Knapen (Reviewer #3).

Essential Revisions:

The reviewers were impressed by the manuscript and data presented. However, there are two issues that should be addressed in a revision:

1) More details should be provided in the current manuscript about aspects of the methods and analyses. See in particular, the recommendations from Reviewer #2.

2) The manuscript would be stronger if clearer links could be drawn between the different aspects of the data, i.e. from pupil to gamma to behavior. You should carefully consider the suggestions from Reviewer #3.

Reviewer #2 (Recommendations for the authors):

The authors should give more details about how induced oscillations were extracted. The reader is referred only to a bioarchive paper, but some essential details should also be given in the present manuscript. It is written that subject specific theta, α, β, and γ frequencies were determined using 1/f-removal and subsequent fitting of Gaussians to the stimulus-induced power spectra. This is a bit confusing, and it is unclear if the power is computed with rather wide canonical frequency bands matching to theta, α, β, and γ or with narrow-band Gaussians which are reported e.g. in Figure 3d. Please clarify. If the latter, please describe also how data was averaged within the frequency bands. This holds at least for data in Figure 5. Could the authors also add information of what is the frequency and temporal resolution of this approach. This is particularly relevant to the analysis of γ peak frequencies shown in Figure 3b.

The stimulus power was calculated and averaged over 0.3-1.3 s post-stimulus period from stimulus onset. This was despite the stimuli containing change in the contrast and rotation (target events) in a period of 0.3-2 s with the reported mean response time being 484ms. From this description it seems that the analysis period for oscillatory power includes the static grating stimuli, its target events (changes) together with the motor responses to these and for sometimes also data after the motor responses. Maybe there is something missing from the Methods section to this description. If not, then the reported γ band responses contain various different forms of neuronal activity some of which cannot be related to behavior nor stimulus processing and which the data-analysis in its present form.

Is these data reported and analysis performed at sensor level or at the source level or is source level data only used for plotting the relevant brain areas?

It is not clear whether data in all panels from Figure 3, that is the main figure of the paper, is group averaged data or data from single subjects. This should be specified for each panel as it is not clear in the current version.

The authors report stimulus specificity of the gamma-band repetition suppression. Can this be due to transient increase in task difficulty by new block of stimuli rather than being stimulus repetition effect per se? The increased feed-back connectivity suggest that this might be the case.

Reviewer #3 (Recommendations for the authors):

This is great work. I have no big comments, but am very interested to see more fleshing out of the possible relations between the different types of data, and how they jointly shed light on the topic of interest.

Due to technical reasons and experimental design choices, the direct investigation of the interrelations between the different signals of interest is difficult; Granger causality analyses could only be performed on those trials for which no pupil-MEG γ correlation exists, and although behavior showed similar time-courses to MEG γ their correlation was not significant. This leaves the link between the different signals up in the air. I would like to suggest a set of analyses that might elucidate this a bit more.

The first of these are concerned specifically with the pupil size analysis.

In my hands, there is a strong inverse correlation between slow and fast pupil responses. The constrictions that the authors here use, are likely joined by slower dilatory changes in the pupil size, making me doubt their purely luminance-based explanations of some of the pupil size signatures. These slower pupil signals could provide a separate source of information that may elucidate the nature of the pupil responses and, possibly, their link to the MEG signals. Could the authors provide the baseline pupil size for all trials, much like they showed the per-trial constrictions? This is because in reversal learning experiments, there is a large pupil dilation when contexts change (cf. doing:10.1371/journal.pone.0185665) and a similar thing might be happening here, too. Such a dilatory response could impact the within-trial stimulus-induced constrictions the authors report. Similarly, the beginning of the experiment is often characterised by a similar slow dilation (cf doi:10.1371/journal.pone.0155574).

Then there's the possibility of blinks either adding large amounts of variance to the pupil size signal, or even that blinks correlate with the experimental block transitions. It would be good if the authors could check this.

Then, the specific time-course analyses. Why 10 trials? This seems a bit arbitrary from how the manuscript is now written. Also, like the authors already say, this pattern of attenuation of signals across blocks hints at two adaptational processes occurring simultaneously. Could the authors not fit two exponential processes, then? The resulting quantification of integration timescales might be an opportune quantification of the signals at hand, and aid in their comparison.

Some questions about the MEG quantifications and results;

– gamma band responses were taken up to 0.6 s post-stimulus-onset (line 829), but these then include the change in contrast of the stimulus in some trials. How do the authors make sure this doesn't impact the results?

– The inflated brain gamma band depictions seem to show ROI boundaries. Is this just an illusory coincidence? Or are there actual differences between ROIs that show themselves in this way?

---

## [Author Response]

Reviewer #2 (Recommendations for the authors):The authors should give more details about how induced oscillations were extracted. The reader is referred only to a bioarchive paper, but some essential details should also be given in the present manuscript. It is written that subject specific theta, α, β, and γ frequencies were determined using 1/f-removal and subsequent fitting of Gaussians to the stimulus-induced power spectra. This is a bit confusing, and it is unclear if the power is computed with rather wide canonical frequency bands matching to theta, α, β, and γ or with narrow-band Gaussians which are reported e.g. in Figure 3d. Please clarify. If the latter, please describe also how data was averaged within the frequency bands. This holds at least for data in Figure 5.Could the authors also add information of what is the frequency and temporal resolution of this approach. This is particularly relevant to the analysis of γ peak frequencies shown in Figure 3b.

We followed the reviewer’s request and extended the relevant Methods section to read:

“Subject-specific theta, alpha, beta, and gamma peak frequencies were determined using 1/f-removal (by fitting and subtracting a linear fit to the semilog power spectrum) and subsequent fitting of Gaussians to the per-participant average stimulus-induced power spectra at the driven V1/V2 dipole (Haller et al., 2018). This procedure found a subject-specific gamma peak for all subjects. If no clear subject-specific theta/alpha/beta peak could be found, a representative peak frequency of the other subjects (theta: 6 Hz, alpha: 10 Hz, beta: 20 Hz) was taken instead.”

“Power over all frequencies was computed for 1 s baseline (-1 s to stimulus onset, power averaged within blocks) and stimulus (0.3 s to 1.3 s post-stimulus onset) data periods. These periods were cut into 50% overlapping windows. For the analysis of frequencies up to 20 Hz, we used 500 ms window length resulting in a spectral resolution of 2 Hz; for frequencies above 20 Hz, we used 333 ms windows, resulting in 3 Hz resolution. The windowed data were demeaned and detrended, then zero-padded to 1 s. For frequencies below 20 Hz, they were Hann tapered, for frequencies above 20 Hz, they were Slepian-window multitapered using three tapers to achieve ±3 Hz smoothing. Finally, the data were Fourier transformed.

For the time-frequency plot in Figure 2D, the same analysis was run, except with zero-padding to 4 s and with windows centered on each time point (at 1200 Hz temporal resolution) between -1 s to 1.3 s relative to stimulus onset.

To quantify power changes at subject-specific theta, alpha, beta and gamma peak frequencies, we used the power-change value at the individual peak frequency bin of the respective band (see Data analysis; Haller et al., 2018).”

“The approach used to estimate trial-averaged theta/alpha/beta/gamma peak frequencies (see Data analysis) could not be used on a per-trial basis, because individual trial spectra were too noisy to estimate the full number of parameters. Therefore, gamma-peak frequency per participant and per trial was estimated as follows: The spectra of per-trial power change relative to baseline were fit with a Gaussian function and, if the fit Gaussian was of positive amplitude and showed a location term between 2-120 Hz, the location term was taken as the per-trial γ peak frequency. “

The stimulus power was calculated and averaged over 0.3-1.3 s post-stimulus period from stimulus onset. This was despite the stimuli containing change in the contrast and rotation (target events) in a period of 0.3-2 s with the reported mean response time being 484ms. From this description it seems that the analysis period for oscillatory power includes the static grating stimuli, its target events (changes) together with the motor responses to these and for sometimes also data after the motor responses. Maybe there is something missing from the Methods section to this description. If not, then the reported γ band responses contain various different forms of neuronal activity some of which cannot be related to behavior nor stimulus processing and which the data-analysis in its present form.

Data after the stimulus change was excluded, and we revised the methods to clarify:

“Data were cut into epochs from 1 s before stimulus onset up to the stimulus change. Trials with stimulus changes before 1.3 s after stimulus onset, trials with missing/early responses, and catch trials were removed. Data segments containing SQUID jumps, muscle artifacts, and blinks were labeled as artifacts. Artifact-free parts of the respective epochs were used for the analyses described below if they contained data for the full respective analysis window lengths, which was the case for 76% of trials.”

Is these data reported and analysis performed at sensor level or at the source level or is source level data only used for plotting the relevant brain areas?

All reported data and performed analyses are in LCMV- or DICS-projected source space, unless explicitly stated, such as in Figure 3 —figure supplement 1. We have added a clarification to the Methods to say so:

“Analyses at the subject-specific theta-, alpha-, beta-, and gamma-band peaks used source projection by means of Dynamic Imaging of Coherent Sources (DICS) beamformers (Gross et al., 2001). All other analyses used source projection by means of Linearly Constrained Minimum Variance (LCMV) beamformers (Van Veen et al., 1997) and were run in source space, except for Figure 3 —figure supplement 1E-F, which explicitly shows sensor level results.”

It is not clear whether data in all panels from Figure 3, that is the main figure of the paper, is group averaged data or data from single subjects. This should be specified for each panel as it is not clear in the current version.

We have revised the figure legend to clarify:

All results in this figure show averages over all participants.

We also clarified in the methods:

All plots show averages over subjects, no single-subject examples are shown.

The authors report stimulus specificity of the γ-band repetition suppression. Can this be due to transient increase in task difficulty by new block of stimuli rather than being stimulus repetition effect per se? The increased feed-back connectivity suggest that this might be the case.

We agree with the reviewer that the early gamma-power decreases might well be driven by increased difficulty/alertness during the first few presentations of each stimulus, and we claim stimulus-specificity only for the late increases. We have found a few instances where this was not clear from our text, and have changed “changes” to “increases” at these lines. We have also explicitly added task difficulty to our paragraph interpreting the early gamma power decrease:

“Unexpected stimuli, or transient increases in task difficulty created by them, likely engage mechanisms of attention and/or arousal, which can be gauged by measuring pupil size. Pupil diameter has been linked to arousal in several studies (de Gee et al., 2017; Peinkhofer et al., 2019). Pupil dilation can best be used in studies that avoid changes in stimulus luminance. Paradigms including luminance increases, as used here, induce pupil constrictions, referred to as the pupillary light reflex, which can also be influenced by arousal, attention and stimulus novelty (Binda et al., 2013; Naber et al., 2013). In our data, pupil constrictions were strong on initial stimulus presentation, decreasing over the first ten repetitions and remaining low for further repetitions. Gamma power showed correlated dynamics for the initial 10 presentations of a given stimulus, but showed increases for further repetitions. This is consistent with a scenario in which stimulus novelty leads to strong gamma and pupil responses for the initial presentation of a stimulus and the rapid decline thereafter, and other mechanisms lead to the steady increase in γ for later repetitions.”

Reviewer #3 (Recommendations for the authors):This is great work. I have no big comments, but am very interested to see more fleshing out of the possible relations between the different types of data, and how they jointly shed light on the topic of interest.Due to technical reasons and experimental design choices, the direct investigation of the interrelations between the different signals of interest is difficult; Granger causality analyses could only be performed on those trials for which no pupil-MEG γ correlation exists, and although behavior showed similar time-courses to MEG γ their correlation was not significant. This leaves the link between the different signals up in the air. I would like to suggest a set of analyses that might elucidate this a bit more.The first of these are concerned specifically with the pupil size analysis.In my hands, there is a strong inverse correlation between slow and fast pupil responses. The constrictions that the authors here use, are likely joined by slower dilatory changes in the pupil size, making me doubt their purely luminance-based explanations of some of the pupil size signatures. These slower pupil signals could provide a separate source of information that may elucidate the nature of the pupil responses and, possibly, their link to the MEG signals. Could the authors provide the baseline pupil size for all trials, much like they showed the per-trial constrictions? This is because in reversal learning experiments, there is a large pupil dilation when contexts change(cf. doing:10.1371/journal.pone.0185665) and a similar thing might be happening here, too. Such a dilatory response could impact the within-trial stimulus-induced constrictions the authors report. Similarly, the beginning of the experiment is often characterised by a similar slow dilation (cf doi:10.1371/journal.pone.0155574).

We thank the reviewer for the pointers to very relevant literature on pre-stimulus pupil dilation. We have added a supplementary figure showing the baseline pupil size for all trials (Figure 3 —figure supplement 1C).

“(C) Pupil size during the trial baseline (-0.3 to 0 s), on a per-trial basis. Baseline pupil size showed a slow decrease after the beginning of the experiment, but no changes at block boundaries. Values in (A‑C) were z-scored within subjects and then averaged over subjects. “

We do not see clear changes in baseline pupil size at block boundaries, neither visually nor in an application of our standardized regression model on prestimulus pupil size. We added the following text to the results (and the corresponding analysis and results to the online code):

“The changes in pupil constriction over repetitions were not driven by changes in pre-stimulus pupil size: Pre-stimulus pupil size did not change over the first ten stimulus repetitions (p=0.15) nor with further stimulus repetitions (p=0.59, Figure 3 —figure supplement 1C).”

As predicted by the reviewer, and as described in the provided literature (Knapen et al., 2016), we see a strong many-trial decrease in pupil dilation at the beginning of the experiment. We previously interpreted these changes to be driven by slow adaptations to the high projector luminance and thank the reviewer for drawing our attention to this compelling alternative explanation. We have added the alternative interpretation demonstrated in Knapen et al. (2016) to the methods:

“As pre-stimulus pupil size was strongly decreasing at the beginning of the experiment (Figure 3 —figure supplement 1C and Figure 4 —figure supplement 1A), possibly due to effects on tonic pupil size of starting an experimental task (Knapen et al., 2016), or due to slow adaptation to the bright light of the projector, pupil size data from the first block and its repetition block were removed from all pupil size analyses.”

As we already exclude the first block from all pupil size analyses due to this strong decrease, and as the pre-stimulus pupil size is stable after the first block, the reported within-trial stimulus-induced constrictions are likely unaffected by slow changes in pre-stimulus pupil size.

Then there's the possibility of blinks either adding large amounts of variance to the pupil size signal, or even that blinks correlate with the experimental block transitions. It would be good if the authors could check this.

To prevent blinks during the periods used for MEG data analysis, and to minimize variance induced by blinks, we only started our experiment after we successfully trained our participants to blink during each inter-trial interval and to prevent blinking during the trial. Blinks therefore occurred during merely 3.9% of all trials (data and code have been added to online code repository). We have added text to the methods to specify so:

“Subjects were trained before the experiment to avoid eye blinks during the baseline and stimulation period and to instead blink during each inter-trial interval. Therefore, blinks occurred in merely 3.9% of all analyzed trials.”

We have also added a new supplementary figure panel that shows the average number of blinks per trial (z-scored within subjects, then averaged over subjects). As can be seen, there seems to be no consistent pattern, neither at block transitions nor overall (Figure 3 —figure supplement 3B).

“(B) Per-trial number of blinks during the trial interval, on a per-trial basis, z-scored within subjects and averaged over subjects. For plots A and B, average and 95% bootstrap confidence intervals were computed using a five-trial-wide running window within each block.”

The low number of blinks prevented our standardized multi-level linear regression model (as well as simplified forms of it) from converging when being applied to the number of blinks per trial.

We considered a fixed-effect version of the regression model that pools trials over participants to guarantee convergence. We computed such a simplified generalized linear model (number of blinks assumed to be Poisson distributed and linearly modeled using a log link) with number of blinks per trial as dependent, and repetition number, trial number and block changes as independent variables. There were no significant effects of any of the three predictors (p = 0.08, p = 0.10, and p = 0.87, respectively). We would, however, caution against interpreting this model too strongly, as the pooling of trials over participants likely renders it anticonservative. Ultimately, the low number of blinks recorded here prevents in our view a sensible statistical analysis. We have therefore refrained from reporting this analysis and merely report it here for the attention of the reviewer.

Then, the specific time-course analyses. Why 10 trials? This seems a bit arbitrary from how the manuscript is now written. Also, like the authors already say, this pattern of attenuation of signals across blocks hints at two adaptational processes occurring simultaneously. Could the authors not fit two exponential processes, then? The resulting quantification of integration timescales might be an opportune quantification of the signals at hand, and aid in their comparison.

We originally attempted to fit the per-participant data as the sum of one exponential decrease and a linear or log-linear increase, but found the data on the per-subject level to be too noisy to reliably estimate time constants for individual subjects. As a more robust alternative, we chose linear fits and the cut-off at 10 trials, which was chosen using a broken stick regression.

We revised the results to clarify:

“Across stimulus repetitions, gamma showed a biphasic pattern that could be well described by a linear decrease until a breakpoint at the 10^th^ repetition, followed by a linear increase (see Methods).”

We added the following text as the corresponding methods part:

“On the source level, we fit per-repetition band power with two regression lines: one line from trial 1-10, and a separate line from trial 11-120. To find this breakpoint between early gamma power decrease and late gamma power increase, we averaged per-repetition gamma power over the four non-repeated blocks, z‑scored within each participant and averaged over participants. The resulting subject-average time course over repetitions was smoothed with a moving average over 10% of all repetitions and a broken line fit was applied. The best-fitting breakpoint was found to be repetition 10.”

We additionally followed the suggestion of the reviewer, leading to the following addition to the methods and results. In the methods, we added:

“To estimate the time constants τ of early gamma power and pupil constriction decreases, we averaged per-repetition γ power and pupil constriction over the four non-repeated blocks, z‑scored within each subject and averaged over subjects. The resulting subject-average time courses over repetitions were fit as the sum of an exponential and a linear process over repetitions, y=aexτ+bx+c. Resulting fit values were Radj2=0.68 for γ power and Radj2=0.36 for pupil constriction.”

In the Results on gamma power, we added:

“We also fitted the pattern of gamma over all repetitions with the sum of an exponential decay and a linear increase and found the early decrease to have a time constant of 3.5 repetitions (CI95%=[2.34.6])”

In the Results on pupil constriction, we added:

“As for gamma, we also fitted the pattern of pupil constriction over all repetitions with the sum of an exponential decay and a linear increase and found the early decrease to have a time constant of 5.6 repetitions (CI95%=[2.68.7]).”

Some questions about the MEG quantifications and results;– Gamma band responses were taken up to 0.6 s post-stimulus-onset (line 829), but these then include the change in contrast of the stimulus in some trials. How do the authors make sure this doesn't impact the results?

Data after the stimulus change was excluded, and we revised the methods to clarify:

“Data were cut into epochs from 1 s before stimulus onset up to the stimulus change. Trials with stimulus changes before 1.3 s after stimulus onset, trials with missing/early responses, and catch trials were removed. Data segments containing SQUID jumps, muscle artifacts, and blinks were labeled as artifacts. Artifact-free parts of the respective epochs were used for the analyses described below if they contained data for the full respective analysis window lengths, which was the case for 76% of trials.”

– The inflated brain Gamma band depictions seem to show ROI boundaries. Is this just an illusory coincidence? Or are there actual differences between ROIs that show themselves in this way?

We assume that the reviewer is referring to our area labeling: We missed to specify that the area labels introduced in Figure 2A are also overlaid in Figures 2E and figures 3E-F to ease assignment of the reported effects to cortical areas and to notify the reader of the focus of most effects on the early visual cortices. To make this clear, we added the following sentence to all figure legends of figures using the cortical area shading:

“Black-to-white shading indicates areas V1, V2, V3, V3A, and V4.”